# Fetch and Forge: Efficient Dataset Condensation for Object Detection

**Ding Qi**[1][*]   **Jian Li**[2][†]   **Jinlong Peng**[2]   **Bo Zhao**[4]
**Shuguang Dou**[1]   **Jialin Li**[2]   **Jiangning Zhang**[2]
**Yabiao Wang**[3,2][†]   **Chengjie Wang**[2]   **Cairong Zhao**[1][†]
[1]Tongji University [2]Tencent YouTu Lab
[3]Zhejiang University [4]Shanghai Jiao Tong University

## Abstract

Dataset condensation (DC) is an emerging technique capable of creating compact synthetic datasets from large originals while maintaining considerable performance. It is crucial for accelerating network training and reducing data storage requirements. However, current research on DC mainly focuses on image classification, with less exploration of object detection. This is primarily due to two challenges: (i) the multitasking nature of object detection complicates the condensation process, and (ii) Object detection datasets are characterized by large-scale and high-resolution data, which are difficult for existing DC methods to handle. As a remedy, we propose DCOD, the first dataset condensation framework for object detection. It operates in two stages: Fetch and Forge, initially storing key localization and classification information into model parameters, and then reconstructing synthetic images via model inversion. For the complex of multiple objects in an image, we propose Foreground Background Decoupling to centrally update the foreground of multiple instances and Incremental PatchExpand to further enhance the diversity of foregrounds. Extensive experiments on various detection datasets demonstrate the superiority of DCOD. Even at an extremely low compression rate of 1%, we achieve 46.4% and 24.7% $AP_{50}$ on the VOC and COCO, respectively, significantly reducing detector training duration.

## 1   Introduction

In the past decade, the field of deep learning has witnessed the emergence of high-performance models, driven by the development of convolutional and Transformer networks [10, 24, 8, 5]. These models, while benefiting from large-scale datasets, also encounter significant challenges such as data storage and the demand for extensive training resources. Dataset condensation (or distillation) [27, 32, 2] has emerged in response, aiming to synthesize a small amount of data that approximates the training effectiveness of original datasets, thus offering hope to alleviate this dilemma.

Current dataset condensation (DC) research mainly focuses on image classification and is divided into two main frameworks: Meta-learning and Data-matching [11]. As shown in Figure 1 (a), the Meta-learning framework [27, 17, 34, 15] optimizes synthetic datasets by minimizing performance risks on the original dataset's validation set, using a bi-level optimization process. In contrast, Figure 1 (b) illustrates the Data-matching framework, which aligns gradients [32], feature distributions [31], or training trajectories [2] to simulate the original data's impact during different model training stages.

---

[*]Work done when Ding Qi is an intern in Tencent YouTu Lab, and the advisor is Jian Li.
[†]   Corresponding author

38th Conference on Neural Information Processing Systems (NeurIPS 2024).

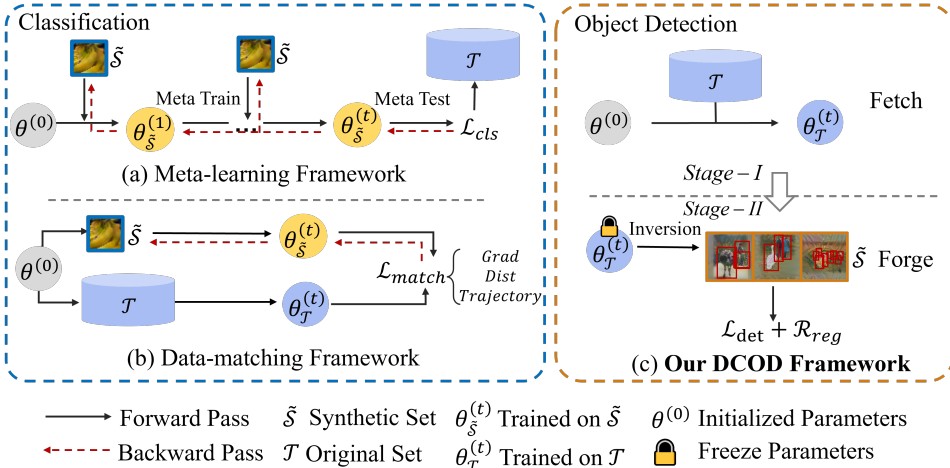

**Figure 1:** (a) Meta-learning treats synthetic set updates in DC as a meta-task; (b) Data-matching indirectly matches gradients, feature distributions, and training trajectories generated during network training; (c) Our proposed DCOD, the first for object detection, decouples the bi-level optimization of methods a and b, following a Fetch and Forge two-stage approach.

While both frameworks above make significant contributions to the field, they share common limitations. The complexity and computational expense of their bi-level optimization process restrict their application, typically confining them to smaller datasets (e.g., CIFAR10, CIFAR100, Tiny-ImageNet) and simpler network architectures (e.g., ConvNet-3, AlexNet, VGG-11, ResNet-18). This limitation is particularly pronounced in object detection tasks, which are inherently more complex than image classification. Specifically, i) the requirement in object detection for simultaneous localization and classification of multiple objects in an image considerably complicates the condensation process; and ii) the large-scale, high-resolution nature of detection dataset images, along with the complexity of detection network structures, poses significant challenges in extending existing DC methods.

In this study, we propose DCOD, **the first dataset condensation framework for object detection.** DCOD is distinct in its ability to flexibly generate foreground objects of varying sizes, shapes, and categories at any position within an image, as demonstrated in Figure 2. Additionally, DCOD streamlines the process by eliminating the complexities of traditional bi-level optimization, enhancing its compatibility with complex detection networks and large-scale, high-resolution data. **DCOD operates in two stages: Fetch and Forge.** As in Figure 1 (c), in the first stage Fetch, we train an object detector on the original dataset following a standard procedure, during which key information from the original data is stored within the detector. Then, in the second stage, we employ the model inversion to synthesize images from the trained detector. Specifically, we freeze the detector's weights and input initialization images and labels (including position, size, and category) randomly sampled from the original dataset to capture the instance distribution of real originals. During the inversion process, we propose the Foreground Background Decoupling ($\mathcal{F}_{BD}$) to enhance attention to the foreground areas through random erasure guided by a coarse mask and Incremental PatchExpand ($\mathcal{I}_{PE}$) expands a single image into multiple patches, each guided by different target labels, thus synthesizing a richer variety of instances. Finally, we utilize the detector's loss function as the guiding task loss for dataset condensation. To ensure the quality of the generated images, we implement two types of regularization: pixel-level alignment and feature-level alignment. Our method underwent rigorous evaluation on the MS COCO [13] and Pascal VOC [7, 6] datasets, achieving top-tier results that highlight its significant effectiveness. With some problems left open and a considerable improvement room existing, we hope this pilot study will attract more community interest.

The primary contributions of this paper can be summarized as follows:

- We propose DCOD, the first dataset condensation framework for object detection datasets. Utilizing a two-stage process of Fetch and Forge, it can simultaneously condense crucial localization and classification information from the original dataset.

- For the multi-object distribution characteristic of object detection datasets, we propose the Foreground Background Decoupling strategy and Incremental PatchExpand which notably boost the diversity of multi-instances, all within a constrained storage budget.

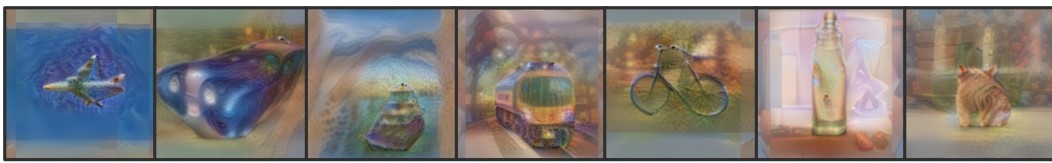

(a) Single-object synthetic images

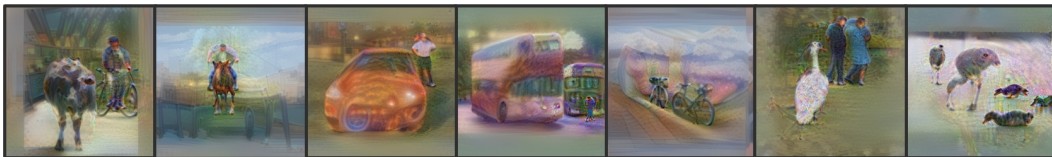

(b) Mutliple-object synthetic images

**Figure 2:** Visualization of images synthesized using DCOD. (a) A synthetic image contains only one category of foreground. (b) A synthetic image contains multiple foregrounds of different sizes, shapes, and categories.

- Extensive experiments on various detection datasets underscore the superiority of DCOD. Remarkably, even at an extremely low compression rate of 1%, we achieve 46.4% and 24.5% AP50 on the VOC and COCO datasets respectively, significantly reducing the training duration for object detection.

## 2 Related work

**Dataset Condensation.** Dataset condensation (or distillation) aims to compress large-scale original data into a small amount of synthetic data, accelerating network training while maintaining comparable performance. Current research primarily focuses on the field of image classification and can be divided into two frameworks: meta-learning and data matching. In the meta-learning framework, methods such as DD [27], KIP [17], RFAD [15], and FRePo [34] update model parameters in the inner loop and synthetic data in the outer loop. In the data matching framework, techniques like DC [32], DSA [30], and IDC [9] utilize gradient matching, comparing neural network weight gradients from training on both real and synthetic data. Distribution matching approaches, exemplified by DM [31], CAFE [26], DataDAM [22], and IDM [33], align real and synthetic data distributions using Maximum Mean Discrepancy (MMD), often through a single optimization process but potentially limiting the performance. Trajectory matching, such as MTT [2], optimizes synthetic data by pre-calculating and storing training trajectories of expert networks from the real dataset. A critical downside of MTT is the significant storage requirement for these trajectories.

Recent research suggests compressing original data into models rather than synthetic data. DiM [25] minimizes the difference in predicted logits between real and generated images, using a generator to store original dataset information. SRe$^2$L [29] indicates that key dataset information can be preserved in deep neural network training. Inspired by this, we argue that the localization and classification information of the detection data can be learned and preserved by the detector, allowing the key information to be recovered to reconstruct the synthetic image. Unlike SRe$^2$L, we focus on object detection datasets with multiple instances in a single image. We propose a foreground-background decoupling strategy and incremental PatchExpand to enhance the updating of synthesized multiple instances.

Diverging from the traditional focus on image classification, our study pioneers dataset condensation in object detection. We introduce a streamlined approach that separates the traditional bi-level optimization into a two-stage Fetch and Forge process. This strategy effectively realizes dataset condensation in object detection tasks with increased efficiency.

**High Training Overhead in Object Detection.** Object detection methods are broadly divided into two categories: one-stage and two-stage detectors. One-stage detectors, such as YOLO series [19, 20], SSD [14], DSFD [12], are valued for their speed and simplicity. Conversely, two-stage detectors like Faster R-CNN [21] prioritize accuracy. Training object detectors, particularly on a single GPU, demands substantial time. Training a complex model like Faster R-CNN on datasets such as COCO can take several days or even weeks, depending on the configuration and hardware. This extensive training time underscores the pressing need for more efficient training methodologies in object

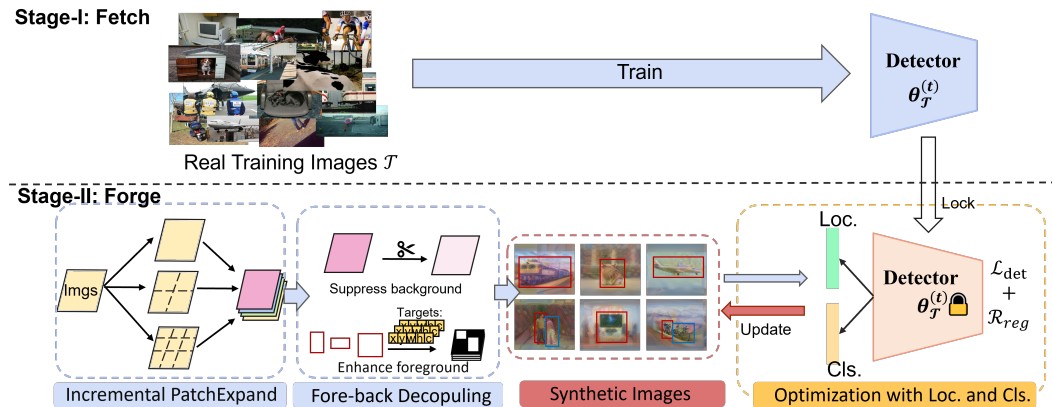

**Figure 3:** Overview of the DCOD framework: In the first stage, Fetch, a detector is trained on the original images, encapsulating key information from the original dataset. In the second stage, Forge, a randomly initialized synthetic set is enhanced through Foreground Background Decoupling and Incremental PatchExpand on the initial images, which are then input into the trained detector. Guided by targets, specific category targets are updated in the corresponding areas of the images. The loss of the detector serves as the task loss for condensation, while pixel-level and feature-level regularization ensure the quality of the generated images.

detection, highlighting the growing importance of dataset condensation advancements. To address this issue, we introduce the first dataset condensation framework for object detection.

## 3 Dataset Condensation for Object Detection

### 3.1 Preliminary

Dataset condensation reduces a large real dataset $\mathcal{T}$ into a smaller synthetic dataset $\widetilde{\mathcal{S}}$. We first extend the existing DC [27, 32] optimization framework to accommodate detection task. For object detection, each image is annotated with multiple bounding boxes and their associated class labels. The real dataset $\mathcal{T}$, containing $N$ labeled images, is given by $\mathcal{T} = \{(X_i, A_i)\}_{i=1}^N$, where $A_i = \{(x_j, y_j, w_j, h_j, c_j)\}_{j=1}^{n_i}$, with $x_j, y_j, w_j, h_j$ representing the coordinates and sizes of the $j$-th bounding box, and $c_j$ being the class label. The synthetic dataset $\widetilde{\mathcal{S}}$, comprising $M$ images where $M \ll N$, is formulated as $\widetilde{\mathcal{S}} = \{(\widetilde{X}_i, \widetilde{A}_i)\}_{i=1}^M$, where $\widetilde{A}_i = \{(\widetilde{x}_j, \widetilde{y}_j, \widetilde{w}_j, \widetilde{h}_j, \widetilde{c}_j)\}_{j=1}^{\widetilde{n}_i}$ represents the synthetic bounding boxes and their corresponding class labels. The optimization process for dataset condensation is twofold:

Model Parameters Update (Inner Loop): The model parameters $\theta$ are optimized over the synthetic dataset $\widetilde{\mathcal{S}}$ to minimize the loss function $L^{\widetilde{\mathcal{S}}}(\theta)$:

$$\theta_{\widetilde{\mathcal{S}}}(\widetilde{\mathcal{S}}) = \arg\min_\theta L^{\widetilde{\mathcal{S}}}(\theta). \tag{1}$$

Synthetic Dataset Update (Outer Loop): The synthetic dataset $\widetilde{\mathcal{S}}$ is refined to minimize the loss on the real dataset $\mathcal{T}$ using the model parameters $\theta_{\widetilde{\mathcal{S}}}(\widetilde{\mathcal{S}})$ obtained from the inner loop:

$$\widetilde{\mathcal{S}}^* = \arg\min_{\mathcal{S}} L^{\mathcal{T}}(\theta_{\widetilde{\mathcal{S}}}(\widetilde{\mathcal{S}})). \tag{2}$$

However, this optimization process underlying this procedure involves a complex bi-level optimization scheme, which leads to high computational costs. The challenge intensifies when dealing with intricate model structures in the inner-loop or large synthetic datasets in the outer-loop. In the context of object detection, where high-resolution images and complex network structures are the norm, employing such bi-level optimization becomes even more impractical. This emphasizes the need for more streamlined and practical methods in dataset condensation, especially for tasks like object detection that involve high computational demands.

## 3.2 Fetch and Forge: DCOD Framework

Pioneering the application of dataset condensation to object detection, we develop the Dataset Condensation for Object Detection (DCOD) framework, which is a novel two-stage process depicted in Figure 3. This method diverges from the traditional bi-level optimization that compresses information directly into synthetic data. Instead, we optimize the model and synthetic data separately. Subsequent sections will delve into the optimization objectives for the Fetch and Forge stages.

**Stage-I: Fetch.** For object detection tasks, the real dataset is given by $\mathcal{T} = \{(X_i, A_i)\}_{i=1}^{N}$, with $A_i$ encompassing bounding boxes and class labels. By training an object detector $\psi_\theta$ parameterized with $\theta$ on the original dataset $\mathcal{T}$, we capture key information crucial for the tasks of localization and recognition within images. The optimization can be expressed as:

$$\theta_\mathcal{T} = \arg \min_\theta \mathcal{L}_{\text{det}}(\psi_\theta(X), A), \tag{3}$$

with $\mathcal{L}_{\text{det}}$ representing the composite loss function that combines localization loss $\mathcal{L}_{\text{loc}}$ and classification loss $\mathcal{L}_{\text{cls}}$:

$$\mathcal{L}_{\text{det}}(\psi_\theta(X), A) = \mathcal{L}_{\text{loc}}(\psi_\theta(X), A) + \mathcal{L}_{\text{cls}}(\psi_\theta(X), A), \tag{4}$$

minimizing $\mathcal{L}_{\text{det}}$ adjusts $\theta$ to detect objects with higher precision.

**Stage-II: Forge.** In the forge stage, we borrow principles from model inversion [16, 28, 3], utilizing a well-trained detection model to guide the embedding of essential information back into the synthetic images. To achieve this, it is essential to recognize the unique aspects of detection tasks: an image typically contains multiple foreground objects with different positions, sizes, and shapes. Therefore, we propose Foreground Background Decoupling ($\mathcal{F}_{BD}$) and Incremental PatchExpand ($\mathcal{I}_{PE}$). Finally, we use the detection loss as the task loss to guide the update of the targets' corresponding areas with the foregrounds of various categories.

**Foreground-Background Decoupling.** We initialize the synthetic set $\widetilde{\mathcal{S}} = \{(\widetilde{X}_i, \widetilde{A}_i)\}_{i=1}^{M}$ by randomly selecting a subset of images and their corresponding targets from the original dataset, which includes position coordinates and category labels. The primary objective of object detection is to distinguish the foreground from the background and then classify the foreground objects. Therefore, we prioritize synthesizing the foreground. Additionally, while the background is treated as a single category in detection tasks, we adopt a background suppression strategy to avoid blending different contextual semantics (e.g., integrating sky or ocean information into a grassy background).

To this end, we propose Foreground-Background Decoupling ($\mathcal{F}_{BD}$) to separately handle the updating of the foreground and background areas, as follows:

$$\mathcal{F}_{BD}(\widetilde{X} = \{x_{\text{back}}, x_{\text{fore}}\}) = \begin{cases} x_{\text{back}} \leftarrow \alpha x_{\text{back}} & (\alpha < 1) \\ \mathcal{R}_E(x_{\text{fore}}) \end{cases} \tag{5}$$

We first separate the foreground and background regions using a binary mask based on the bounding box coordinates from the original labels. For the background ($x_{\text{back}}$), we apply a suppression strategy controlled by a hyperparameter $\alpha$, which limits updates to preserve contextual information. For the foreground ($x_{\text{fore}}$), we employ a random erasure method $\mathcal{R}_E$, ensuring the model focuses more on refining foreground pixels during the inversion phase. By strategically applying suppression and enhancement, this approach prioritizes important visual information, optimizing the model's ability to generate high-fidelity synthesized images.

**Incremental PatchExpand.** Unlike classification datasets that contain only one object, images in detection datasets sometimes contain multiple instances of several different categories. Inspired by curriculum learning, we propose Incremental PatchExpand ($\mathcal{I}_{PE}$) to learn more instances during training. By adopting this approach, we increase the number of divisions, thereby introducing complexity into the synthetic data. We first standardize all images to a uniform size via padding. Subsequently, we employ a procedure, as defined by the function:

$$\mathcal{I}_{PE}(\widetilde{X}) = \text{Expand}(\text{Patch}(\text{Padding}(\widetilde{X}), k), \widetilde{A}) \tag{6}$$

---

**Algorithm 1** Dataset Condensation for Object Detection

---

**Input:** Training set $\mathcal{T}$

**Required:** Randomly initialized set of synthetic samples $\widetilde{\mathcal{S}}$, initialized detector $\psi_\theta$ parameterized with $\theta$, training iterations K, learning rate $\eta_s$

  1: **Stage-I:** Fetch
  2: Train $\psi_\theta$ on $\mathcal{T}$: $\theta_\mathcal{T} = \arg\min_\theta \mathcal{L}_{\det}(\psi_\theta(X), A)$
  3: **Stage-II:** Forge
  4: **for** $k = 0, \ldots, K - 1$ **do**
  5:     Apply $\mathcal{F}_{BD}$ and $\mathcal{I}_{PE}$ for synthetic set $\widetilde{\mathcal{S}} = \mathcal{F}_{BD}(\mathcal{I}_{PE}(\widetilde{\mathcal{S}}))$
  6:     Compute task loss $L_{det}$ using Equ. 4
  7:     Compute $R_{\text{reg}}$ using Equ. 8, where compute $R_{pixel}$ and $R_{feature}$ using Equ. 9 and Equ. 11
  8:     Calculate $L = L_{\det}(\psi\theta_\mathcal{T}(\widetilde{X}), \widetilde{A}) + R_{reg}$
  9:     Update $\widetilde{\mathcal{S}} \leftarrow \widetilde{\mathcal{S}} - \eta_s \nabla_s L$
10: **end for**

**Output:** Condensed synthetic dataset $\widetilde{\mathcal{S}}$

---

which divides the images into $k \times k$ patches and each patch is then optimized for different targets, enhancing the diversity in object position, size, and shape. The incremental introduction of more patches allows the model to gradually adapt to various complexities and learn effectively across a broader range of scenarios.

**Optimization with Pixel and Feature Regularization.** Following the model inversion and decoupling framework [28, 29], the synthetic images $\widetilde{X}$ are optimized by solving the following minimization problem:

$$\widetilde{X} = \underset{\widetilde{X}}{\arg\min} \, L_{\det}(\psi_{\theta_\mathcal{T}}(\widetilde{X}), \widetilde{A}) + R_{reg}, \tag{7}$$

where $L_{\det}$ denotes the detection loss and $\theta$ signifies the parameters of the model applied to $\widetilde{X}$. Here, $\widetilde{A}$ represents the set of annotations for the targets present in $\widetilde{X}$. The term $R_{reg}$, a regularization term, is crucial for preserving the intrinsic qualities of the original dataset within the synthetic images and is defined as the sum of pixel and feature regularization:

$$R_{reg} = R_{pixel} + R_{feature}, \tag{8}$$

The pixel-level regularization $R_{pixel}$ comprises two terms:

$$R_{pixel}(\widetilde{X}) = \alpha_{TV} R_{TV}(\widetilde{X}) + \alpha_{l_2} R_{l_2}(\widetilde{X}). \tag{9}$$

with $R_{TV}$ promoting spatial smoothness through the Total Variation regularization:

$$R_{TV}(\widetilde{X}) = \sum_{i,j} \sqrt{(\widetilde{X}_{i,j} - \widetilde{X}_{i+1,j})^2 + (\widetilde{X}_{i,j} - \widetilde{X}_{i,j+1})^2}, \tag{10}$$

and $R_{l_2}$ minimizing the squared Euclidean norm of pixel values. The coefficients $\alpha_{TV}$ and $\alpha_{l_2}$ are hyperparameters that balance the Total Variation and $L_2$ terms, respectively.

Feature-level regularization $R_{feature}$ aims to align the feature statistics of the synthetic image $\widetilde{X}$ with the batch normalization (BN) layers of the pre-trained model, as follows:

$$R_{feature}(\widetilde{X}) = \sum_l \left( \|\mu_l(\widetilde{X}) - \mu_l^{BN}\|_2^2 + \|\sigma_l^2(\widetilde{X}) - \sigma_l^{BN}\|_2^2 \right), \tag{11}$$

where $\mu_l(\widetilde{X})$ and $\sigma_l^2(\widetilde{X})$ represent the mean and variance of the activations at the $l$-th BN layer of $\widetilde{X}$, while $\mu_l^{BN}$ and $\sigma_l^{BN}$ correspond to the statistics derived from the pre-trained model. This feature regularization is fundamental for the consistency of the feature distribution and the utility of $\widetilde{X}$ in training object detectors. At last, we summarize the complete two-stage process in Algorithm 1.

Table 1: The Performance comparison results on Pascal VOC, with the compression ratios of 0.5%, 1%, and 2%. The methods compared include Random [18], Uniform, K-Center [23] and Herding [1, 4]. The models used in both the compression and evaluation phases are YOLOv3-SPP. Ratio (%): the ratio of condensed images to the whole training set.

| Methods | mAP (%) | $AP_{50}$(%) | $AP_{75}$(%) | $AP_s$(%) | $AP_m$(%) | $AP_l$(%) |
|---|---|---|---|---|---|---|
| Ratio 0.5 (%) | | | | | | |
| Random | 4.9±0.2 | 15.8±0.7 | 1.4±0.1 | 0.2±0.1 | 1.4±0.1 | 6.7±0.4 |
| Uniform | 5.0±0.2 | 15.8±0.5 | 1.5±0.1 | 0.1±0.1 | 1.5±0.1 | 6.7±0.3 |
| K-Center | 3.8±0.3 | 14.5±0.6 | 0.9±0.1 | 0.01±0.0 | 0.7±0.12 | 5.1±0.4 |
| Herding | 3.6±0.0 | 12.6±0.2 | 0.9±0.01 | 0.1±0.1 | 0.9±0.1 | 4.6±0.1 |
| Ours | **14.2±0.5** | **37.9±0.9** | **6.6±0.4** | **2.4±0.3** | **6.6±0.4** | **18.1±0.7** |
| Ratio 1 (%) | | | | | | |
| Random | 8.6±0.5 | 25.5±0.7 | 3.0±0.2 | 0.4±0.1 | 3.1±0.1 | 12.6±0.6 |
| Uniform | 8.8±0.4 | 25.7±0.6 | 3.2±0.3 | 0.4±0.1 | 3.2±0.2 | 12.8±0.4 |
| K-Center | 6.1±0.2 | 21.9±0.9 | 1.4±0.1 | 0.1±0.0 | 1.3±0.2 | 8.3±0.3 |
| Herding | 5.5±0.2 | 19.3±0.5 | 1.3±0.1 | 0.2±0.0 | 1.3±0.1 | 7.9±0.3 |
| Ours | **19.8±0.4** | **46.4±0.4** | **13.0±0.6** | **4.4±0.2** | **10.5±0.3** | **24.9±0.5** |
| Ratio 2 (%) | | | | | | |
| Random | 15.8±0.5 | 40.5±0.7 | 8.2±0.7 | 1.4±0.6 | 5.8±0.4 | 20.5±0.4 |
| Uniform | 15.9±0.3 | 40.6±0.5 | 8.2±0.5 | 1.5±0.3 | 5.8±0.3 | 20.6±0.3 |
| K-Center | 9.21±0.2 | 31.3±0.5 | 2.4±0.1 | 0.4±0.1 | 2.5±0.1 | 12.4±0.3 |
| Herding | 8.5±0.4 | 28.1±0.8 | 2.6±0.1 | 0.5±0.1 | 2.4±0.2 | 12.3±0.2 |
| Ours | **23.7±0.6** | **50.7±0.6** | **18.7±0.5** | **5.6±0.7** | **13.6±0.5** | **29.9±0.7** |
| Whole Dataset | 46.5±0.5 | 76.4±0.4 | 46.7±0.3 | 11.7±0.6 | 28.2±0.5 | 52.2±0.4 |

Table 2: The Performance comparison results on MS COCO, with the compression ratios of 0.25%, 0.5%, and 1%. The models used in both the compression and evaluation phases are YOLOv3-SPP.

| Methods | mAP (%) | $AP_{50}$(%) | $AP_{75}$(%) |
|---|---|---|---|
| Ratio 0.25 (%) | | | |
| Random | 3.5±0.1 | 9.7±0.1 | 1.6±0.1 |
| Uniform | 3.6±0.1 | 9.8±0.2 | 1.6±0.1 |
| K-Center | 1.7±0.1 | 6.3±0.0 | 0.4±0.0 |
| Herding | 1.7±0.1 | 5.8±0.1 | 0.5±0.1 |
| Ours | **7.2±0.1** | **17.2±0.2** | **4.8±0.2** |
| Ratio 0.5 (%) | | | |
| Random | 5.5±0.1 | 14.2±0.1 | 2.9±0.1 |
| Uniform | 5.6±0.1 | 14.3±0.1 | 2.9±0.1 |
| K-Center | 2.8±0.0 | 8.9±0.1 | 0.7±0.1 |
| Herding | 2.6±0.1 | 8.8±0.1 | 0.8±0.1 |
| Ours | **10.0±0.1** | **21.5±0.2** | **8.0±0.1** |
| Ratio 1 (%) | | | |
| Random | 8.3±0.0 | 19.7±0.1 | 5.3±0.1 |
| Uniform | 8.4±0.1 | 19.7±0.2 | 5.4±0.1 |
| K-Center | 4.0±0.1 | 12.9±0.1 | 1.2±0.1 |
| Herding | 4.1±0.1 | 12.5±0.2 | 1.3±0.2 |
| Ours | **12.1±0.1** | **24.7±0.2** | **10.4±0.1** |
| Whole Dataset | 36.1±0.2 | 63.6±0.3 | 36.5±0.2 |

# 4 Experiment

## 4.1 Experiment Setup

The standard evaluation of dataset condensation consists of two steps: first, condensing the original training set into a smaller synthetic dataset; then, training an initialized model with this synthetic data and evaluating it on the original dataset's test set to assess the synthetic data's effectiveness in model training.

**Datasets and Metrics.** Our DCOD method is evaluated on Pascal VOC [7, 6] and MS COCO [13] benchmarks, with image resolution set to 512x512. For Pascal VOC, we merge the trainval sets of VOC2007 and VOC2012 into a single training set with 16,551 images, using the VOC2007 test set for evaluation. MS COCO comprises 80 categories with 118,287 training and 5,000 test images. Both datasets are assessed using standard COCO metrics: mAP (mean Average Precision), $AP_{50}$ (0.5 IoU threshold), $AP_{75}$ (0.75 IoU threshold), and size-specific $AP_s$, $AP_m$, $AP_l$ for performance evaluation. We initialize the network with pretrained backbone weights and train it 5 times on the distilled dataset. The evaluation is conducted on the original test dataset to obtain the $\bar{x} \pm std$.

**Implementation Details.** The initialization of the synthetic set involves random sampling from the original images, adhering to the specified compression ratio. The compression rate in classification tasks is typically set to around 1%. In our task, for VOC, we evaluate at compression rates of 0.5%, 1%, and 2%, while for COCO, we report at 0.25%, 0.5%, and 1%. During the standard image synthesis process, we set the image learning rate at 0.002. The weight for the task loss is set at 1, while $R_{feature}$ is assigned a weight of 0.1. The $\alpha_{TV}$ and $\alpha_{l2}$ is established at 1 and 0.001, respectively. The $\alpha$ in background suppression strategy is set to 0.7. All experiments were performed on a single V100 gpu.

## 4.2 Experimental Results

**Counterpart Methods.** As we are the first dataset condensation methods for object detection, we refer to the comparison settings of general DC in classification [27, 32] and choose three core-set selection methods: Random [18], K-center [23], and Herding [1, 4]. The Random method randomly selects real images from the original training set to form a subset. To adapt the K-center and Herding for object detection tasks, we implement a straightforward modification by using a pre-trained feature extractor (ResNet50) to process the features of entire images and employing the L2 norm to measure distances. Additionally, considering class balance, we also employ the Uniform, which is commonly used in practical applications within the field of object detection.

Table 3: Ablation Study on the components of Forge, $\mathcal{F}_{BD}$, and $\mathcal{I}_{PE}$ at 1% compression rate on Pascal VOC, with the model using YOLOv3-SPP.

| Methods | Ratio 1 (%) | | |
|---|---|---|---|
| | mAP(%) | AP$_{50}$(%) | AP$_{75}$(%) |
| baseline | 8.3±0.6 | 25.9±0.5 | 2.9±0.3 |
| baseline+$\mathcal{F}_{BD}$ | 10.0±0.3 | 27.9±0.4 | 3.4±0.2 |
| baseline+$\mathcal{I}_{PE}$ | 17.7±0.4 | 42.6±0.5 | 11.6±0.4 |
| baseline+$\mathcal{F}_{BD}$+$\mathcal{I}_{PE}$ | 19.8±0.4 | 46.4±0.4 | 13.0±0.6 |

Table 4: Cross-architecture performance on Pascal VOC and MS COCO. We use the one-stage detector YOLOv3-SPP and the two-stage detector Faster RCNN for the evaluation phase, respectively.

| Ratio (%) | Detector | VOC | | COCO | |
|---|---|---|---|---|---|
| | | mAP(%) | AP$_{50}$(%) | mAP(%) | AP$_{50}$(%) |
| 0.5 | YOLOv3-SPP | 14.2±0.5 | 37.9±0.9 | 10.0±0.1 | 21.5±0.2 |
| | Faster-RCNN | 6.3±0.4 | 18.5±0.7 | 3.7±0.3 | 8.4±0.2 |
| 1 | YOLOv3-SPP | 19.8±0.4 | 46.4±0.4 | 12.1±0.1 | 24.7±0.2 |
| | Faster-RCNN | 13.8±0.5 | 33.2±0.6 | 6.1±0.4 | 13.5±0.3 |

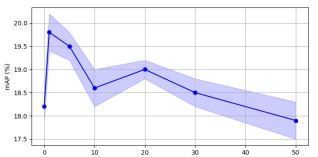

**Figure 4:** Ablation of $\alpha_{TV}$ on Pascal VOC, with a compression ratio of 1%, using YOLOv3-SPP.

Table 5: Comparison of different initialization on Pascal VOC with a compression ratio of 0.5% using YOLOv3-SPP.

| Initial Methods | mAP(%) | AP$_{50}$(%) |
|---|---|---|
| Noise | 6.9±0.7 | 23.2±0.5 |
| Random | 14.2±0.5 | 37.9±0.9 |
| K-center | 11.9±0.6 | 34.2±0.4 |
| Herding | 11.5±0.7 | 33.8±0.6 |

**Comparison Results.** Table 1 and 2 provide the comparative results of our dataset condensation method on the VOC and COCO datasets, underlining its superior performance at all compression rates. According to the official ultralytics/yolov3 implementation, the detector achieves an mAP of 46.5% and AP$_{50}$ of 76.4% on VOC when trained with the full dataset, and 35.6% mAP on COCO, which serves as an approximate upper limit of performance. On VOC, using a 2% compression rate, our method still reaches an AP$_{50}$ of 50.7%. On COCO, at a 1% compression rate, the mAP and AP$_{50}$ reach 12.1% and 24.7% respectively. Compared to random sampling, at the lowest compression rate of 0.5% on VOC, our method increases mAP and AP$_{50}$ by 9.3% and 22.1% respectively, and at COCO's lowest compression rate of 0.25%, our method improves mAP and AP$_{50}$ by 3.7% and 7.5% respectively. Moreover, while the Uniform method shows slight improvement over Random, it still falls short of the performance achieved by our approach. The Coreset method, focusing only on the overall image features and neglecting instance-level details, shows poor performance. Our method also demonstrates comprehensive advantages in multi-size object detection metrics like AP$_s$, AP$_m$, AP$_l$, indicating that our DCOD significantly enhances the diversity of foreground positions, sizes, and shapes.

### 4.3 Ablation Study

**Effectiveness of Each Component.** Table 3 shows that using model Inversion from Forge as our baseline, our proposed augmentations $\mathcal{I}_{PE}$ and $\mathcal{F}_{BD}$ improve performance by 1.7% and 9.4% respectively. Combined, they yield an 11.5% overall improvement. Notably, the baseline alone underperforms compared to random sampling. We found that without these augmentations, the detection loss quickly converges in early iterations, missing key positional and classification details. Thus, incorporating both $\mathcal{I}_{PE}$ and $\mathcal{F}_{BD}$ is crucial to enhance the diversity and effectiveness of the synthetic images.

**Cross-architecture Generalization.** We verify the performance of condensed data on new architectures, and in Table 4, our approach is able to maintain good generalization at 1% compression. However, we observe a larger performance drop at lower compression rates. This is due to the fact that the one-stage detector and two-stage detector architectures differ significantly, preserving limited information when too few images are synthesized.

**Sensitivity of Hyper-Parameters.** Figure 4 shows how different $\alpha_{TV}$ regularization weights affect our method's performance, using the VOC dataset with a 1% compression rate. The results indicate that an optimal regularization weight enhances performance, while no regularization or too much regularization reduces it. This happens because lack of regularization introduces excessive noise, and too much regularization causes excessive smoothing in the synthetic images, both of which degrade the quality of the images.

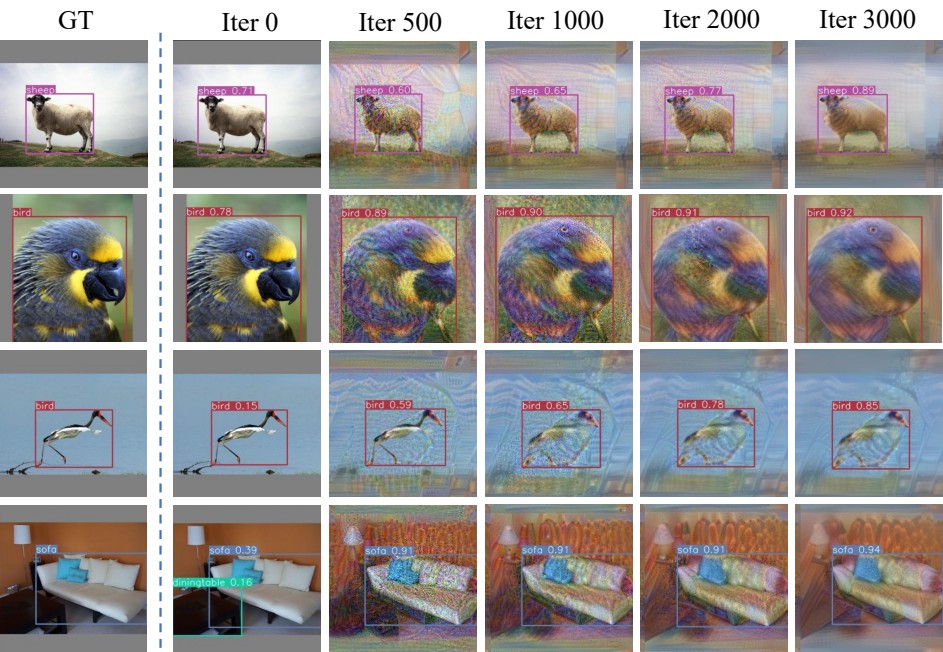

**Figure 5:** Visualization of different iteration steps during the condensed phase. Scores are assigned using the trained YOLOv3-spp model, based on IOU@0.5. The "GT" column represents the true labels of images. "iter0" shows the initial image scores, followed by scores of synthetic images and their bounding box at iterations 500, 1000, 2000, and 3000.

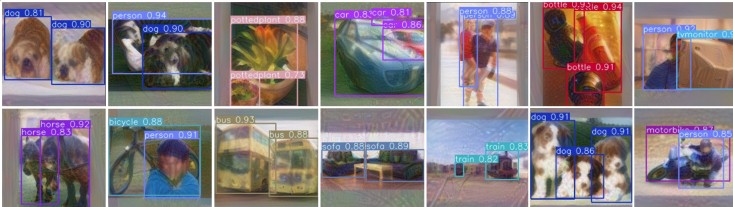

**Figure 6:** Visualization of the multi-instance synthesized images. Scores are assigned using the trained YOLOv3-SPP model, based on IOU@0.5.

**Performance upper bound**   As shown in Figure 7, we compare the performance variations of the Random baseline method and the DCOD method as the compression ratio increases. The performance of the full dataset, marked by a gray line, serves as the theoretical upper bound. When the ratio is below 5%, DCOD shows a significant advantage over the random method, while as the ratio exceeds 20%, the performance of both methods converge.

**Discussion on Initialization**   We discuss the impact of initialization sampling on the performance of the synthetic dataset in this section. As shown in Table 5, the Random method, which is the primary choice for the experiments in this paper, is simple and performs well. Initialization strategies based on K-center and Herding lead to a decline in performance. This is these core-set methods fail to account for the complex factors in detection datasets, such as class distribution and object size distribution, resulting in significant differences between the synthetic and original datasets. Using noise as the initialization method, due to the absence of any prior information from the original dataset, causes a notable drop in the effectiveness of the synthetic dataset.

### 4.4  Visualization

We present a comparative analysis of synthetic images across various iterations, as shown in Figure 5. Notably, there is a progressive enhancement in the foreground scores with advancing iterations, indicative of the incremental integration of category-specific beneficial information into the images. Specifically, for sheep in the first row, the scores at iter500 and iter1000 start lower than the initial

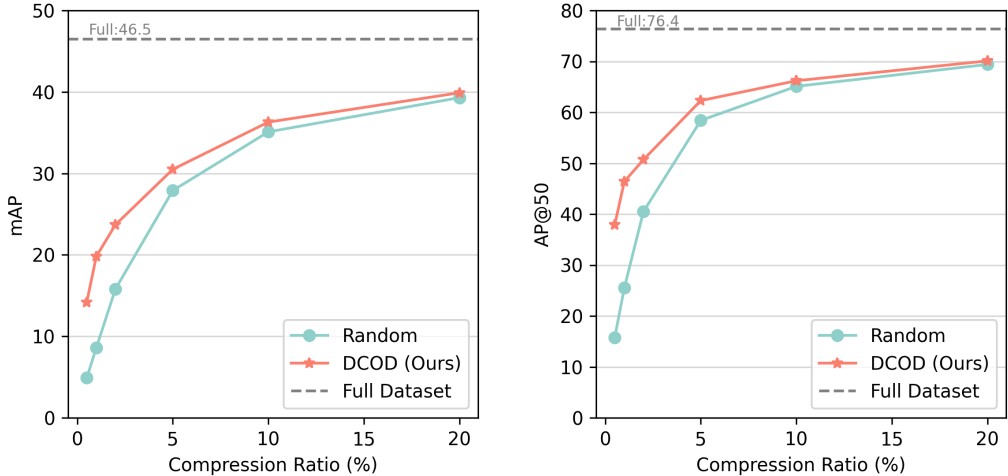

**Figure 7:** Visualization of performance curves on Pascal VOC using YOLOv3-SPP as the compression ratio increases.

image but surpass initial scores after iter2000. This trend implies that early images are initially disrupted by noise from the update process, but as iterations progress, pixel, and feature regularization help maintain image clarity.

Additionally, we also showcase the visualization results of synthetic images containing multiple instances, as shown in Figure 6, demonstrating that DCOD effectively supports the generation of multiple instances required in detection tasks.

## 5   Conclusion, Limitations and Future Work

In this study, we introduce the first dataset condensation framework for object detection, DCOD. It is a two-stage framework where, in the first stage, Fetch stores information vital for detection from the original dataset into model parameters. In the second stage, Forge, the images are then reconstructed through model Inversion. We propose foreground background decoupling and incremental Patch-Expand, to improve the efficiency and diversity of the synthetic images. Extensive experiments are conducted on two standard detection benchmarks, Pascal VOC and MS COCO, to demonstrate that DCOD can significantly maintain superior performance even at an extremely low compression rate.

We acknowledge the limitations of our work from two perspectives. First, due to the significant differences among object detectors, we did not extend our approach to more complex detectors like DETR, as this might require more specialized designs to accommodate their structures. Second, the performance across different architectures is still insufficient and remains an area that needs improvement.

## Acknowledgments and Disclosure of Funding

This work was supported by National Natural Science Fund of China (62076184 , 62473286, 62306046) in part by Shanghai Natural Science Foundation (22ZR1466700).

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
