# OpenReview forum: "Fetch and Forge: Efficient Dataset Condensation for Object Detection"
_NeurIPS.cc/2024/Conference — NeurIPS 2024 poster_

### Official Review · Reviewer_SZRX · 2024-06-27

**Soundness:** 3
**Presentation:** 3
**Contribution:** 3
**Rating:** 5
**Confidence:** 4

**Summary:**

1) The paper introduces a dataset compression technique and applies it for the first time in object detection tasks.
 2) For large-scale object detection tasks, the condensation of datasets can not only shorten model training time but also save a lot of computational resources.
3) The method proposed in the paper is simple and effective, which can effectively solve the problems raised.

**Strengths:**

DCOD is distinct in its ability to flexibly generate foreground objects of varying sizes, shapes, and categories at any position within an image. Additionally, DCOD streamlines the process by eliminating the complexities of traditional bi-level optimization, enhancing
 its compatibility with complex detection networks and large-scale, high-resolution data.

**Weaknesses:**

1.Algorithm needs to be optimized.
2.The description of the second stage of DCOC is not clear enough.

**Questions:**

1、In the experimental section, it is possible to add several more datasets for comparison of condensing methods.
2、Why is the author's two-stage model framework more effective than a complex bidirectional compression optimization framework? Please discuss in detail.
3、It cannot be seen from Figure 3 what the target in the annotation refers to.
4、Formula 5's introduction is not specific enough.
5、How is each patch optimized for different goals and increased complexity in the IPE module?
6、Some annotations are missing in Tables 1 and 2, such as how the results of the entire dataset are obtained. What does "±" mean?
7、Based solely on the experimental results, due to the lack of comparison with similar experiments, the experimental results lack some persuasiveness.
8、How is the synthesized image updated under the guidance of loss?Suggest adding composite images with multiple instances in the visualization result graph.
9、Lack of reference to Figure 5 in the visualization section.

**Limitations:**

1.Due to the significant differences among object detectors, you did not extend your approach to more complex detectors like
 DETR, as this might require more specialized designs to accommodate their structures.
2. The performance across different architectures is still insufficient and remains an area that needs improvement.

---

> ### Author Rebuttal · Authors · 2024-08-07
>
> ### Q1 [Algorithm needs to be optimized]
> In this work, we analyze why the traditional bidirectional framework is challenging for detection tasks (lines 35-40). We propose the fetch and forge two-phase framework, which avoids the computational cost of second-order derivatives and achieves critical optimization for detection tasks. Due to the complexity of detection tasks, further detailed optimizations will be explored in future research.
> ### Q2 [The description of the second stage is not clear enough]
> For unclear areas, please refer to the specific responses provided in the following questions. We will include more details in the revised version.
> ### Q3 [Add several more datasets.]
> We understand the reviewer's concerns and will consider adding larger-scale general detection datasets, such as Object-365. However, due to time constraints, completing these experiments in the short term is challenging. If the paper is accepted, we will include additional detection datasets in the revised version.
> Moreover, Pascal VOC and MS-COCO are the two most influential benchmark datasets in the object detection field. As far as we know, many well-known detection papers have also only used these two datasets. We have demonstrated the effectiveness of our method on these datasets.
> ### Q4 [Discuss about two-stage framework vs bidirectional framework ]
> The bidirectional method optimizes outer image updates by comparing performance differences between real and synthetic sets, using gradients or model parameters to compute matching loss. The model also requires inner loop optimization involving gradient calculations, resulting in second-order derivatives and high computational costs. As image resolution and model complexity increase, the efficiency of second-order derivative calculations decreases, making outer loop optimization for synthetic images challenging. As stated in lines 32-35, these methods are only effective on small datasets and networks.
> Our two-stage method decouples the original second-order calculations, allowing separate training of a complex detector and optimization of high-resolution images (lines 43-45). This separation improves efficiency.
> ### Q5 [Figure 3 lack target.]
> Thank you for pointing that out. The term "target" refers to the objects randomly sampled along with the images during initialization. We will supplement the targets in Figure 3 in the revised version.
> ### Q6 [Add formula 5's introduction.]
> We are pleased to provide a more detailed explanation:
> In synthesizing the image $\widetilde{X}$, we separate the foreground and background regions using a binary mask. For the background ($x_{\text{back}}$), we apply a suppression strategy with a hyperparameter $\alpha = 0.7$, limiting updates to preserve contextual meaning. For the foreground ($x_{\text{fore}}$), we employ a random erasure method, ensuring that the model focuses more on refining the foreground pixels during the inversion phase. By strategically applying suppression and enhancement, we prioritize the processing of important visual information, optimizing the model's ability to generate high-fidelity synthesized images.
> ### Q7 [Patch optimization and goal complexity in IPE module.]
> We employ an incremental approach to enhance the model's adaptability to an increasing number of ground truth labels and bounding boxes. Initially, we introduce a modest number of gt labels and bboxes, allowing the model to assimilate the initial dataset without overwhelming its learning capacity. Our methodology progresses through three stages, each comprising 1000 iterations. Within each stage, we incrementally increase the instance count, denoted by k, starting from 1*1 and progressing to 2*2, and finally to 3*3. This curated incrementation ensures a robust and sustainable learning trajectory.
> ### Q8 [Missing annotations in Table1&2 and "±" explanation. ]
> Thank you for pointing out this issue. The performance on the full dataset is based on the open-source implementation from official ultralytics/yolov3. The "±" symbol represents the standard deviation calculated from 25 results, obtained by generating five synthetic sets and testing each set five times.
> ### Q9 [Lack of comparison with similar methods]
> To our knowledge, our work is the first to study dataset condensation for object detection. Traditional methods for dataset condensation in classification are difficult to apply to detection tasks, making direct comparisons challenging.
> However, we understand the reviewer's concerns and are considering adding a discussion section to analyze the difficulties and challenges of adapting classification dataset condensation pipelines to detection tasks. This discussion could inspire future research in this area.
> In fact, early papers on classification dataset condensation (such as [Dataset Distillation] and [Dataset Meta-Learning from Kernel Ridge-Regression]) only compared against random and core-set methods. Therefore, we are essentially following their experimental framework.
> ### Q10 [How is the synthesized image updated under the guidance of loss? add multiple instances in the visualization.]
> Please refer to Figure 3 in the supplementary PDF, where we show the variation in guidance loss over 3000 iterations and the test performance of synthetic images(tested every 100 iterations).
> We also add more visualization for multi-instance in Figure 4 of the supplementary PDF.
> ### Q11 [Lack of reference to Figure 5.]
> Apologies for the oversight. We will add a reference to Figure 5 in the revised paper. Thank you for pointing it out.
> ### Q12 [ Limitation ]
> The two limitations are actually discussed in the "Future Work" section of our paper.
> 1.We will investigate more complex detector architectures following the inversion approach.
> 2.We plan to build a model pool of different detectors during the fetch phase and perform inversion using multiple detector weights on the synthestic image in the forge phase to incorporate diverse information.

---

> ### Author Response · Authors · 2024-08-13
> **Gentle Reminder**
>
> Dear Reviewer SZRX,
>
> We hope our response addressed your questions. As the discussion period comes to a close, we eagerly await your valuable feedback. We would greatly appreciate it if you could consider improving your rating.
>
> Thank you for your valuable time!
>
> Sincerely,
>
> Authors of Paper #8952

---

### Official Review · Reviewer_8kYh · 2024-07-05

**Soundness:** 3
**Presentation:** 3
**Contribution:** 3
**Rating:** 7
**Confidence:** 4

**Summary:**

The paper attempts to generalize dataset condensation to object detection. It proposes Fetch and Forge. Fetch: training a normal object detector on the original dataset as usual. Forge: the synthetic images are sampled from the original data and optimized by the detection loss. Experiments on VOC and COCO show the effectiveness of the method.

**Strengths:**

1. The paper is well-motivated and easy to understand.

2. The proposed methods are technically sound.

3. The experiments show promising results and there is much room for future.

**Weaknesses:**

1. Notation issue: see $\mathcal{I}_{PE}$ in Eqn. 6 and Step 5 of Alg.1.

2. Why is row 3 of Table 3 blank?

3. Implementation Details are not comprehensive. Is it the training epoch for IPE set to $M$/batchsize? And I would suggest not calling it a training epoch.

4. What would happen if we increase the condensation ratio, e.g., 5%, 10%, 20%, 50%?

5. Typos: it should be ''ratio'' in Table 3.

**Questions:**

see weaknesses above

**Limitations:**

see weaknesses above

---

> ### Author Rebuttal · Authors · 2024-08-07
>
> ### Q1 [Notation issue: see IPE in Eqn. 6 and Step 5 of Alg.1.]
> We note the inconsistency in the notation for I_PE between Equation 6 and Step 5 of Algorithm 1. We will correct this inconsistency in the revised version of the paper. Thank you for bringing this to our attention.
> ### Q2 [Why is row 3 of Table 3 blank?]
> The blank row in Table 3 was due to a LaTeX formatting error. It should have displayed the results for the "baseline+IPE" method. We will correct this issue in the revised version of the paper. Thank you for pointing it out.
> ### Q3 [Implementation Details are not comprehensive. Is it the training epoch for IPE set to M/batchsize? And I would suggest not calling it a training epoch.]
> Yes, we set M/batchsize for IPE. We will correct the misleading terminology by using "iterations" instead of "epochs." Thank you for the suggestion, and we will include more detailed implementation details in the revised version.
> ### Q4 [What would happen if we increase the condensation ratio, e.g., 5%, 10%, 20%, 50%?]
> Based on your suggestion, we add experiments on VOC with compression ratios of 5%, 10%, 20%, and 50%. See tabel:
>
> |  VOC Rate |   Methods    |    mAP      |     AP50    |
> |-----------| ------------ | ----------- | ----------- |
> |      5%   |    Random    |    27.9     |     58.4    |
> |      5%   |    Ours      |    30.5     |     62.3    |
> |     10%   |    Random    |    35.1     |     65.1    |
> |     10%   |    Ours      |    36.3     |     66.2    |
> |     20%   |    Random    |    39.3     |     69.4    |
> |     20%   |    Ours      |    39.9     |     70.1    |
> |     50%   |    Random    |    41.5     |     72.6    |
> |     50%   |    Ours      |    41.7     |     72.8    |
>
> We find that as the ratio increases, the advantage of the condensation method compared to the baseline gradually decreases, which is consistent with some studies in classification([DC-BENCH: Dataset Condensation Benchmark]). In future work, exploring how to train using only synthetic images to surpass the performance upper bound of real images is a very interesting topic.
> ### Q5 [it should be ''ratio'' in Table 3.]
> Thank you for pointing out the spelling error. "Ratio" refers to the compression ratio of the entire dataset. We will correct this in the revised version of the paper.

---

> > ### Comment · Reviewer_8kYh · 2024-08-11
> >
> > Thanks for the response. Most of my concerns are addressed.
> > I have one question on Q3.
> > Please describe the training details for Stage-II: Forge.

---

> ### Author Response · Authors · 2024-08-11
>
> ### Stage-II: Forge Phase
> We are pleased to provide a more detailed explanation:
>
> **Initialization**: Randomly sample original images and their corresponding targets from the real dataset to form the initial synthetic set $\widetilde{\mathcal{S}}$, using the detector $\psi_{\theta_\\mathcal{T}}$ trained in Stage-I (with weights frozen).
>
> **Training (Inversion)**: The optimization goal is to update the pixels of the synthetic images.
>
> 1. Divide $\widetilde{\mathcal{S}}$ into $k \times k$ patches using the IPE module. The FBD module constructs a binary mask to differentiate between foreground and background, applies random erasure to the foreground, and sets the update weight for background pixels to $\alpha=0.7$.
> 2. Input $F_{BD}$($I_{PE}$($\widetilde{\mathcal{S}}$)) into the detector, and calculate the loss according to Equation 7 in the paper: $L = L_{\text{det}}(\psi_{\theta_{\mathcal{T}}}(\widetilde{X}), \widetilde{A}) + R_{\text{reg}}$.
> 3. Update the pixels: $\widetilde{\mathcal{S}} \leftarrow \widetilde{\mathcal{S}} - \eta_s \nabla_s L$.
>
> Under the guidance of the task loss, the synthetic images gradually stabilize, as shown in Figure 3 and 4 of the supplementary PDF.

---

> ### Comment · Reviewer_8kYh · 2024-08-11
>
> Thanks for the quick reply. So, is the training iteration 3K? As shown in Fig. 3 of the supplementary PDF, mAP goes for 94+ at 3000 iterations. While in the main paper, there are no such experimental settings that can exceed 94 mAP.
> So I wonder what is the experimental setup corresponding to this figure. Also, is there any learning rate decay?

---

> ### Author Response · Authors · 2024-08-11
>
> We appreciate your attention to detail.
>
> Yes, the number of training iterations is 3,000.
>
> In Figure 3, we recorded the synthetic images within a batch, as indicated by "mAP for current Batch" on the right column of the images. This mAP was calculated by directly predicting on the current batch of synthetic images using a pre-trained YOLOv3-SPP model. However, in the experiments described in the paper, we trained a detector from scratch on the full synthetic set and then tested it on the original test set. Thank you for your reminder; we will include all the specific details of Figure 3 in the revised version.
>
> we apply learning rate decay using the `CosineAnnealingLR` scheduler, with `eta_min` set to 0.0005.

---

> > ### Comment · Reviewer_8kYh · 2024-08-11
> >
> > About Q4: What is the mAP when ratio=100%? This may be the upper bound of dataset condensation. I also suggest adding a plot that depicts the mAP growth w.r.t. condensation ratio.

---

> > > ### Author Response · Authors · 2024-08-12
> > >
> > > Ratio = 100% means the whole-dataset training, which is the upper-bound of dataset condensation, the performance is 46.5% mAP. Thanks for the good suggestion! We will add a plot to depict the mAP growth w.r.t. condensation ratio in the final version.

---

### Official Review · Reviewer_8mRE · 2024-07-12

**Soundness:** 3
**Presentation:** 3
**Contribution:** 3
**Rating:** 6
**Confidence:** 5

**Summary:**

The paper looks at the task of dataset condensation for object detection. While there have been many works looking at dataset condensation for classification this is the first work to look at it for detection which is more challenging as each image can contain multiple objects of different categories. This is an important problem as it reduces the training requirements. The proposed method operated in two stages, first training a model on the full data. Followed by model inversion to get synthetic images which capture the most important classes and regions. The authors also propose new techniques to handle the multi object nature of images and show good results on VOC and COCO.

**Strengths:**

The paper has the following strengths:
1. The paper looks at an important and challenging problem of dataset condensation for object detection. I have seen many works look at data distillation for classification but this is more challenging and novel task to look at.
2. The authors provide a clear way to propose the first method for this task. They remove the need for expensive 2 step approaches which include meta learning or feature/data matching and rather simplify the problem with the proposed fetch and forge approach.
3. The authors show results on common datasets like VOC and COCO and while they cannot match the full data performance I think it might be a good first step.
4. The authors do ablation of various components, hyperparams and generalization and also show qualitative results.

**Weaknesses:**

The paper has the following weaknesses:

1. In object detection and recognition multiple works have show that the context (background) plays an important role. [Noise or Signal: The Role of Image Backgrounds in Object Recognition]. The Fetch a Forge paper continuously suppresses background (L155-1156) and considers background not important. Is there backing of these claims and why would context (background) not help?
2. Lacking discussion about the input initialization. What effect do different input initializations have. Is there a way to find an optimal initialization? I think the initialization part needs to be discussed in more detail.
3. Section 3.1 seems less relevant given the paper is talking about object detection not classification. It should be modified to describe the setup in terms of detection data.
4. Missing discussion of what happens with different initializations during the Fetch part. Do you get similar performance and visualization of the condensed data?
5. Missing analysis of whether during condensation for object detection location is more important or classification while performing inversion to get the condensed synthetic samples.
6. The authors should also report at what percent of condensed data they can meet the real number. That is important to get some idea of compression to number of images tradeoff and what the best performance this method can give.

Small clarifications:
1. Its confusing why incremental patch expand is needed if the network is already trained during fetch step.
2. Missing ablation regarding the Reg loss in Eq 7 and also within it the importance of pixel vs feature regularization.
L112 For the complex of -> For the complex case/scenario
L167 course -> coarse

**Questions:**

I think the paper is well written and in general easy to follow and sound. At the same time I have mentioned a few clarification and discussions in the weaknesses and would hope the authors answer some of them. Especially around background, initializations and upper bound with more images. Overall I feel the problem is important and the authors propose a first step towards solving it.

**Limitations:**

Yes.

---

> ### Author Rebuttal · Authors · 2024-08-07
>
> ### Q1 [Background suppression and its impact on object detection.]
> We acknowledge the importance of background in detection tasks but prioritize updating the foreground in image synthesis while retaining the background for better results. "Suppress background" means limiting updates to background pixel values to preserve contextual integrity. For example, a background originally containing the sky could be mixed with information like the ocean or grass, disrupting the original context. We will clarify that background is not "unimportant" and add a discussion on this topic. Our ablation study on 1% VOC shows that increasing background update weight decreases performance, confirming our suppression strategy.
> |fore_weights|back_weights|mAP|AP50|
> |-|-|-|-|
> |1|1|18.6±0.8|44.1±0.7|
> |1|0.7|19.8±0.4|46.4±0.4|
> |1|0.5|19.6±0.5|46.1±0.3|
> |1|1.5|18.1±0.5|43.5±0.8|
> |1|2|17.7±0.3|42.9±0.7|
> ### Q2 [Discussion about the input initialization.]
> In this paper, we randomly sampled images and labels from the real dataset for initialization, leading to variations in the synthetic set's foreground categories and object size distribution. This caused result fluctuations. Defining a good initialization for detection dataset condensation is complex, considering factors like category frequency and bounding box characteristics. We supplemented experiments on VOC 0.5% with different methods, including noise, random, k-center, herding, and a 'size-balance' method suggested by Reviewer #z3Af.
>
> |intial method|mAP|AP50|
> |-|-|-|
> |noise|6.9±0.7|23.2±0.5|
> |random|14.2±0.5|37.9±0.9|
> |k-center|11.9±0.6|34.2±0.4|
> |herding|11.5±0.7|33.8±0.6|
> |Size-balance|14.8±0.4|38.5±0.3|
>
> Noise performed the worst, consistent with experiences in classification dataset condensation. Random and Size-balance achieved good results. Future studies on initialization in the detection domain could be key to improving condensation performance. We will continue exploring how to select representative samples to enhance efficiency and effectiveness.
> ### Q3 [Section 3.1 should be more on detection data]
> Thank you for your suggestion.
> We will revise Section 3.1 to include a detailed description of the setup for the detection dataset condensation task
> ### Q4 [what happens with different initializations during the Fetch part.]
> The Fetch phase involves standard training of detectors, with initialization referring to the detector's initial weights. The weights obtained after Fetch training are also used as initial model weights for the Forge phase, influencing the quality of synthesized images. We tested models trained for different numbers of epochs for inversion, leading to varying performances, as shown in the table below for 0.5% VOC:
>
> |training epoch|mAP|AP50|
> |-|-|-|
> |100|14.1±0.3|37.5±0.5|
> |300|14.2±0.5|37.9±0.9|
> |500|13.8±0.6|37.1±0.6|
> |1000|11.4±0.5|33.1±0.7|
>
> Prolonged Fetch phase training can negatively affect synthesized image quality, possibly due to excessive data compression, complicating reconstruction. This issue needs further investigation in future work.
> ### Q5 [Importance of location vs. classification in condensation]
> We conducted a new experiment to address this concern by adjusting the weights of the classification and location losses during the inversion process. For YOLOv3-SPP, we used GIoU loss for location and CrossEntropyLoss for classification. The results are shown in the table below with a 0.5% compression ratio on VOC.
>
> |CEloss|GIoUloss|mAP|AP50|
> |-|-|-|-|
> |1|1|19.8±0.4|46.4±0.4|
> |1|1.5|19.7±0.6|46.2±0.5|
> |1|2|19.3±0.3|45.9±0.4|
> |1|0.5|19.2±0.4|45.5±0.7|
> |1|0.2|18.3±0.8|44.7±0.6|
>
> We found that a 1:1 weight ratio is very suitable. When the weight for the localization loss is further reduced, the performance metrics show a noticeable decline.
> ### Q6 [Report percentage of condensed data meetting real data.]
> Due to time constraints, we increase the compression ratio on the voc as follows:
>
> |VOCRate|Methods|mAP|AP50|
> |-|-|-|-|
> |5%|Random|27.9|58.4|
> |5%|Ours|30.5|62.3|
> |10%|Random|35.1|65.1|
> |10%|Ours|36.3|66.2|
> |20%|Random|39.3|69.4|
> |20%|Ours|39.9|70.1|
> |50%|Random|41.5|72.6|
> |50%|Ours|41.7|72.8|
>
> We find that as the ratio increases, the advantage of the condensation method compared to the baseline gradually decreases, which is consistent with some studies in classification([DC-BENCH: Dataset Condensation Benchmark]). In future work, we will further explore how to achieve lossless methods.
> ### Q7 [why IPE is needed if the network is already trained]
> Incremental Patch Expand is used in the Forge phase to increase the diversity of the synthetic images. Taking into account the complexity of multi-instance detection task, we introduced the incremental approach to gradually increase the number of instances being optimized.
> ### Q8 [pixel vs feature regularization]
> We have add an ablation study on pixel versus feature regularization on 1%ratio VOC, as shown in the table below:
>
> |Rpixel|Rfeature|mAP|AP50|
> |-|-|-|-|
> |√|√|19.8±0.4|46.4±0.4|
> |×|√|17.8±0.5|43.2±0.3|
> |√|×|1.8±0.2|3.5±0.5|
> |×|×|0.3±0.0|0.8±0.1|
>
> For inversion techniques, Rfeature is crucial for reconstructing the information content in the synthesized images. Rpixel helps maintain performance to some extent by balancing the image pixel distribution and reducing sharp regions in the pixels.

---

> ### Comment · Reviewer_8mRE · 2024-08-12
>
> Thanks for the authors' response it helped clarify my concerns and adding these to the paper will definitely make it stronger. I appreciate the authors' efforts in the rebuttal. I will improve my rating to weak accept.

---

> > ### Author Response · Authors · 2024-08-13
> >
> > Thank you for increasing your score! Your insights have been incredibly helpful, and we are excited to incorporate the changes based on your suggestions into our paper.
> >
> > Thanks again for your support and valuable input!

---

### Official Review · Reviewer_z3Af · 2024-07-17

**Soundness:** 3
**Presentation:** 3
**Contribution:** 3
**Rating:** 6
**Confidence:** 4

**Summary:**

This manuscript introduces, as far as I’m aware, the first method to do object detection dataset condensation. The method accomplishes this by first training a detection model on the original dataset called the Fetch stage, then uses this trained model to synthesize a condensed dataset through model inversion and regularization, called the Forge stage. Two modules are introduced to aid in inversion: Forward Background Decoupling (FBD) and Incremental PatchExpand (IPE). FBD induces separate processing on background and foreground pixels; background pixels are down weighted and foreground pixels are randomly masked. IPE progressively divides images into smaller and smaller patches as training progresses and processes different targets in each of these patches. A total variation regularizer is used to ensure synthesized image smoothness and a batch norm feature regularizer is used to ensure the batch norm mean and variance statistics are similar between real and synthetic images. Results on generated datasets outperforms random subsamples of the larger real dataset.

**Strengths:**

*S1:* As far as I’m aware, this is the first work on dataset condensation for object detection and appears to be performant.

*S2:* Aside from the missing definitions I’ve mentioned in the weaknesses section, this manuscript was clear and well written.

**Weaknesses:**

**Weaknesses:**

*W1:* The distribution of classes varies quite a bit at small/medium/large object sizes for both VOC and MS-COCO. It’s possible that K-Center and/or Herding sampling strategies conform to this distribution, but if not, I expect that this would be a strong baseline and should be compared against.

*W2:* In line 159, it states “the foreground is derived from the bounding box coordinates of the current targets”. This needs to be more precise. How exactly is it derived? Is a union taken over all bounding boxes of the current batch and those pixels used as the foreground?

*W3:* In line 167, it states “inspired by course learning”. Is this meant to be curriculum learning?

*W4:* “Expand” in Eq. 6 is never defined. Are image patches upsampled be the same size as the original image?

*W5:* The rate of increase for $k$ in the IPE module is never provided.

*W6:* Does $\tilde{A}$ stay the same as the images sampled at initialization to run synthesis on? If so, how are these samples selected? This seems like it would drastically affect performance.

*W7:* line 218: What is “the Uniform”? I have worked in object detection for years and this term isn’t something I recognize. A citation here is needed.

*W8:* Specify how error bars were computed. Are these the standard deviation over multiple runs of dataset condensation?



**Additional comments, not affecting my rating:**

*C1:* The description of object detectors in line 99-102 isn’t quite correct. Fast models here are the YOLO series and SSD models. These are fast because they don’t include additional processing for small objects, e.g., an FPN. Faster R-CNN and RetinaNet are more accurate because they include an FPN.

*C2:* On line 222: what open-source implementation is used? This should have a citation.

*C3:* Line 230: “The Coreset method” I assume this is referring to K-Centers, but terminology should stay consistent.

*C4:* It would be interesting to compare dataset condensations across more similar architectures. For instance, it should be easy to train a dataset condensed on RetinaNet or Faster-RCNN with FCOS, ATSS, or TOOD.

*C5:* It would be interesting to see Figure 5 on multi-class images.

**Questions:**

This manuscript is interesting and if the authors address the weaknesses I’ve listed, I’ll be happy to raise my score from a 4 to a 5 or above. The most involved item I’d like to see is W1. This should hopefully be a relatively straightforward calculation followed by evaluation on already trained models. W2-8 are items that need to be added to the manuscript to ensure the method can be reproducible given the manuscript text.

Additionally, I’ve provided additional comments on how to improve the manuscript at the end of the weaknesses section, but these have not factored into my score.

**Limitations:**

The manuscript has addressed any concerns I’d have regarding limitations.

---

> ### Author Rebuttal · Authors · 2024-08-07
>
> ### Q1 [Analyze the class distribution of small/medium/large objects in the K-Center and Herding.]
> Based on our analysis, we compare the distribution of small(area<32x32), medium, and large(area>96x96) objects sampled by the k-center and herding methods with the original dataset, presenting the results as stacked bar charts. Please refer to Figure 1&2 in the supplementary PDF. In Figure 1, for the VOC dataset, we observe that k-center tends to select larger images, while herding includes both medium and large images, with both methods selecting fewer small-sized objects. In Figure 2, for the COCO dataset, K-center's selection is more aligned with COCO's inherent proportions, though some categories show significant differences. This may be due to COCO's more balanced object size distribution. These sampling discrepancies might contribute to the poor performance of the methods.
> Subsequently, we design a new baseline approach, called 'Size-balance':
> 1. Calculate the frequency of occurrence for each category based on the sampled subset's size.
> 2. For each category, sample the corresponding number of small, medium, and large samples according to the original dataset's proportion.
> Finally, we compare this baseline sampling approach with subsets from VOC 1%, and COCO 1%. The results outperform the random baseline; however, our method still surpasses this new baseline. A more comprehensive comparison table will be provided in the revised submission.
> | Sample ratio |     Methods   |      mAP      |     AP50     |
> |-------------| --------------|-------------- | ------------ |
> |    1% voc    | K-center      |    6.1±0.2    |    21.9±0.9  |
> |    1% voc    | Herding       |    5.5±0.2    |    19.3±0.5  |
> |    1% voc    | Size-balance  |    9.1±0.3    |    26.7±0.5  |
> |    1% coco   | K-center      |    4.0±0.1    |    12.9±0.1  |
> |    1% coco   | Herding       |    4.1±0.2    |    12.5±0.2  |
> |    1% coco   | Size-balance  |    8.7±0.2    |    20.3±0.4  |
> ### Q2 [How foreground be derived?]
> The foreground coordinates are sampled from the real dataset annotations. During the initialization phase, we random sample the images and their corresponding targets from the dataset, and the bounding box coordinates provided in the dataset annotations are used to define the foreground regions.
> ### Q3 [course learning->curriculum learning?]
> Yes, the term "course learning" was intended to refer to "curriculum learning." We appreciate your feedback and will make the necessary correction in the revised manuscript. Thank you for pointing this out.
> ### Q4 [Expand: Are image patches upsampled be the same size as the original image?]
> "Expand" in Equation 6 refers to the process during the train/test pipeline where synthestic image are upsampled to a specified size, typically 512x512. We will clarify this in the revised manuscript.
> ### Q5 [The rate of k increase in IPE.]
> The growth setting for k is straightforward; we linearly increase k based on iterations. Specifically, k increases from 1 -> 2 -> 3, with an increment every 1000 iterations.
> ### Q6 [How does $\widetilde{A}$ be smapled?]
> Yes, $\widetilde{A}$ corresponds to the images sampled during initialization. Since we use random sampling, the performance of the synthesized dataset may experience slight variations due to different initializations. We reported this standard deviation in Experiments.
> Additionally, we have supplemented the experiments as follows:the performance of the synthesized dataset on 0.5% VOC when using other sampling methods as initialization, including the new baseline 'size balance' established in W1. The study of initialization strategies could be a promising direction for future research.
> | intial method |       mAP      |     AP50     |
> | ------------- | -------------- | ------------ |
> |    noise      |    6.9±0.7     |   23.2±0.5   |
> |    random     |    14.2±0.5    |   37.9±0.9   |
> |    k-center   |    11.9±0.6    |   34.2±0.4   |
> |    herding    |    11.5±0.7    |   33.8±0.6   |
> | Size-balance  |    14.8±0.4    |   38.5±0.3   |
> ### Q7 [ What is 'the Uniform'?]
> The 'Uniform' refers to uniform class sampling, meaning that each class appears approximately the same number of times. We will add appropriate annotations in the tables to clarify this term.
> ### Q8 [Are these the standard deviation over multiple runs of dataset condensation?]
> Yes, the error bars represent the standard deviation over multiple runs of dataset condensation. Following the standard dataset condensation training and testing pipeline, we performed dataset condensation five times. For each generated synthetic set, we conducted five testing runs, resulting in a total of 25 results. The standard deviation was then calculated based on these 25 outcomes.
> ### Q9 [The description in line 99-102 isn't quite correct.]
> Thank you for your observation. We will revise the description of object detectors in related works.
> ### Q10 [what open-source implementation is used?]
> We follow this open-source implementation:official ultralytics/yolov3
> ### Q11 ['The Coreset method',terminology should stay consistent]
> In lines 213-214, we mentioned that core-set selection methods include random, K-Center, and herding. We will emphasize this point again in line 230 and ensure consistent terminology throughout the paper. Thank you for your suggestion.
> ### Q12 [Train a dataset condensed on other detector.]
> We agree with the suggestion to deploy more detectors for condensing; however, due to time constraints and the significant differences in environments and configurations of different detectors, extensive code modifications are required. We plan to include these detectors in future versions.
> Additionally, we will consider applying this framework to other tasks, such as segmentation, in future work.
> ### Q13 [It would be interesting to see Figure 5 on multi-class images.]
> We have added visualizations of multi-class images. Please refer to Figure 4 in the supplementary PDF for details.

---

> > ### Comment · Reviewer_z3Af · 2024-08-08
> >
> > Q1 & Q6: Glad to see the stronger baseline also improves your method!
> >
> > Q5 & Q8: Please include this information in the revised manuscript. I’d recommend putting it in the Implementation Details section.
> >
> > Q7: No need to add a citation for uniform sampling. This make sense to me now and I’m satisfied as long as this is clear in the paper.
> >
> > Rebuttal PDF Figures: I think your manuscript would benefit by adding these figures in the appendix with appropriate reference to these figures in the main paper.
> >
> > Overall, I’m satisfied and will raise my final score.

---

> > > ### Author Response · Authors · 2024-08-10
> > >
> > > Thank you for the in-depth feedback, which undoubtedly helps us improve the paper. We are grateful for your willingness to raise the final score.

---

### Author Rebuttal · Authors · 2024-08-07

We are grateful to all reviewers for acknowledging our work and providing valuable comments and suggestions.
Common strengths noted:
1.Recognition of our motivation and contributions as pioneers in studying detection dataset condensation.
2.Demonstrated effectiveness of our method on two widely-used benchmarks, VOC and COCO, underscoring the potential for significant advancements in this domain.
3.Clear and comprehensible writing.
Common question noted:
1.Need for further research on performance limits with increased compression ratios.
2.Further investigation required on the initialization issue.

Additionally, in the supplementary PDF, Figure 1 and Figure 2 address Reviewer z3Af's Q1 regarding the baseline sampling strategy and the distribution of small/medium/large sizes. Figure 3 addresses Reviewer SZRX's Q8 on how task loss guides synthetic images updates. Figure 4 addresses the multi-class multi-instance visualization raised by Reviewer z3Af and Reviewer SZRX. Due to time constraints, the standard deviation in all supplementary experiments was obtained by generating twice and testing three times each.

Finally, we commit to addressing all concerns and incorporating the mentioned experiments in the revised manuscript if the paper is accepted.

---

### Decision · Program_Chairs · 2024-09-25

**Decision:**

Accept (poster)

**Comment:**

The paper addresses the dataset condensation task for object detection, a more challenging problem than classification. It introduces a two-stage method, Fetch+Forge, with a clear description and thorough experimental evaluation. Reviewers generally agree this is a novel approach, though concerns about writing clarity and evaluation quality were noted. The final ratings varied from Borderline Accept to Accept. The meta-review confirms that most issues raised have been resolved, particularly for reviewer SZRX. Given the clear method and significant results, the AC recommends accepting the paper.